# Prognostic and Monitoring Value of Circulating Tumor Cells in Adrenocortical Carcinoma: A Preliminary Monocentric Study

**DOI:** 10.3390/cancers12113176

**Published:** 2020-10-29

**Authors:** Giulia Cantini, Letizia Canu, Roberta Armignacco, Francesca Salvianti, Giuseppina De Filpo, Tonino Ercolino, Gabriella Nesi, Mario Maggi, Massimo Mannelli, Pamela Pinzani, Michaela Luconi

**Affiliations:** 1Endocrinology Unit, Department of Experimental and Clinical Biomedical Sciences “Mario Serio”, University of Florence, 50139 Florence, Italy; giulia.cantini@unifi.it (G.C.); letizia.canu@unifi.it (L.C.); francesca.salvianti@unifi.it (F.S.); giuseppina.defilpo@unifi.it (G.D.F.); mario.maggi@unifi.it (M.M.); massimo.mannelli@unifi.it (M.M.); pamela.pinzani@unifi.it (P.P.); 2Université de Paris, Institut Cochin, INSERM, CNRS, F-75014 Paris, France; roberta.armignacco@inserm.fr; 3Careggi University Hospital (AOUC), 50134 Florence, Italy; tonino.ercolino@unifi.it; 4Division of Pathological Anatomy, Department of Clinical and Experimental Medicine, University of Florence, 50134 Florence, Italy; gabriella.nesi@unifi.it

**Keywords:** CTCs, ACC, prognosis, adrenocortical cancer, liquid biopsy, cancer biomarker

## Abstract

**Simple Summary:**

Carcinoma of the cortical region of the adrenal (ACC) is a rare and aggressive cancer often with poor prognosis and limited therapies. For these reasons, tumor markers for early diagnosis and monitoring the therapy and tumor evolution are required. This paper demonstrates in a cohort of 19 patients affected by ACC, that in a simple blood draw (liquid biopsy), different cells associated with the tumor can be found in samples taken before and after surgery. Among them, the number of circulating tumor cells in blood samples taken before surgery can be predictive of the patients’ survival and tumor recurrence, thus contributing valuable information on the tumor, which may contribute to improve patient management and follow up. Further studies on larger cohorts of ACC patients are required to validate this novel finding.

**Abstract:**

Adrenocortical carcinoma (ACC), a rare and aggressive neoplasia, presents poor prognosis when metastatic at diagnosis and limited therapies are available. Specific and sensitive markers for early diagnosis and a monitoring system of therapy and tumor evolution are urgently needed. The liquid biopsy represents a source of tumor material within a minimally invasive blood draw that allows the recovery of circulating tumor cells (CTCs). CTCs have been recently shown to be detectable in ACC. In the present paper, we evaluated the prognostic value of CTCs obtained by size-filtration in a small pilot cohort of 19 ACC patients. We found CTCs in 68% of pre-surgery and in 38% of post-surgery blood samples. In addition, CTC clusters (CTMs) and cancer associated macrophages (CAMLs) were detectable in some ACC patients. The median number of CTCs significantly decreased after the mass removal. Finally, stratifying patients in high and low pre-surgery CTC number groups, assuming the 75th percentile CTC value as cut-off, CTCs significantly predicted patients’ overall survival (log rank = 0.005), also in a multivariate analysis adjusted for age and tumor stage. In conclusion, though preliminary and performed in a small cohort of patients, our study suggests that CTC number may represent a promising marker for prognosis and disease monitoring in ACC.

## 1. Introduction

Adrenocortical carcinoma (ACC) is a rare aggressive and heterogeneous neoplasia, with an evolution strictly depending on early diagnosis, as the five-year survival in the metastatic forms drops down to 0–28% according to the different series [1]. The challenge in the management of advanced adrenocortical carcinoma depends on the paucity of the pharmacological approaches [1], which are limited to mitotane (MTT) alone or combined with etoposide, doxorubicin and cisplatin (EDP) treatment [2], with a variable range of efficacy. This still makes the radical resection of the mass the best therapeutic option for early stage resectable ACC [3]. For these reasons, specific and sensitive markers for early diagnosis and for monitoring the therapy and the tumor evolution are urgently needed.

The blood liquid biopsy represents an incredible source of tumor material obtained by a simple and minimally invasive procedure, such as a blood drawing [4,5]. Among the tumor material recovered, circulating tumor cells (CTCs) have now gained importance in the oncology field as biomarkers of the tumor development [6]. They are the most informative marker of the tumor status and of the invasive/metastatic potential, as they might also contain cells able to trigger the metastatic process [7]. We have previously demonstrated the specific presence of CTCs in ACC, suggesting that they may represent a good marker for this tumor [8]. CTC count has been so far recognized to have a significant diagnostic and prognostic value as an independent factor in some solid tumors [9], particularly in breast cancer [10,11,12,13], where its power as quantitative biomarker of the tumor response to chemotherapy has been suggested [14].

Significant differences in the survival rate between metastatic patients affected by breast, prostate and colon cancer with less than few CTCs in the blood sampling and those showing a higher number of cells have been reported, with a cut-off depending on the tumor and the CTC separation technique [11,15,16], determining an initial focus of the research on the CTC number. The identification of CTCs into the bloodstream represents a technical challenge because of the limited number of tumor cells with respect to blood cells. Among the available methods for CTC detection, size-filtration techniques are relatively simple, providing the fractioning of the whole blood in a single step and preserving the cell integrity for CTC detection and further characterization [17]. In the present paper, in a monocentric preliminary study conducted on 19 ACC patients, we assessed the validity of CTC number obtained by blood filtration as an independent prognostic marker, as well as the potential use of CTCs to monitor the disease evolution and response to therapy during the patients’ follow-up.

## 2. Results

We analyzed a cohort of *n* = 19 patients with ACC diagnosis, where at least four ScreenCell^TM^ cytologic filters had been obtained from independent pre-surgery (PreS) and post-surgery (PostS) blood samples. The patients’ characteristics and clinical data are reported in Table 1.

The filters were stained with May–Grünwald–Giemsa or with hematoxylin and eosin for CTC morphometric analysis, cell identification and count [8], as shown in Figure 1. CTC positivity for steroidogenic factor-1 (SF-1) was detected by immunocytochemistry performed in parallel filters obtained from the same blood sample used to confirm CTC adrenocortical origin (inset in panel D, Figure 1). The stained filters obtained in duplicate from blood samples taken before (time interval: −3/+2 months) and after surgical removal of the ACC mass (time interval: +3–80 months) were analyzed by the referent pathologist. We found detectable CTCs in 15/19 patients (79%) in PreS filters and in 13/19 patients (68%) in at least one PostS filters; moreover, considering the filter analysis—45/66 (68%) of PreS filters and 80/209 (38%) of PostS filters had positive readings.

In addition to single CTC, CTC clusters (named circulating tumor microemboli, CTMs, *n* = 4 patients) and cancer associated macrophages (CAMLs, *n* = 7 patients) were also found in some ACC patients (Figure 1). In 50% of cases, CAML were associated with CTMs (χ^2^ = 5.7, *p* = 0.017).

The median CTC number and interval of PreS- and PostS-CTCs are reported in Table 1 for the whole cohort of ACC patients. The analysis demonstrated a statistically significant decrease in the median number of CTCs after the tumor removal in the whole cohort of patients (Figure 2).

Table 1 also reports the patients’ clinical data and CTC numbers in PreS and PostS samples of patients stratified in low and high PreS CTC count. The cut off value for this stratification was 9.0 and 1.2 CTCs isolated from 3 mL of blood, corresponding to the values of the 75th percentile non-parametric distributions of the CTC number, respectively, in PreS and PostS samples. The decrease in CTC median after surgery was statistically significant for both low and high PreS CTC groups (Figure 2).

CTC count differed among patients and at the follow-up. In some patients receiving only MTT therapy—chosen for the multiple blood sampling and filters obtained (≥5)—the behavior of CTC count during the follow-up was associated with MTT response and tumor recurrence (Figure 3A–C).

A statistically significant negative association was found between MTT and CTCs in the three patients considered, where MTT levels and CTCs were evaluated in blood samples drawn on the same day (Figure 4).

We then assessed the predictive power of the CTC number obtained in blood samples drawn before the tumor mass removal. Statistical analysis between CTC number in PreS samples and all ACC clinical parameters of ACC (stage, tumor diameter, secretion, Weiss score, stage, number of mitosis, Ki67%) showed a significant positive correlation only with the Weiss score (R = 0.537, *p* = 0.032) and with the metastatic stage IV (R = 0.517, *p* = 0.028).

In order to identify factors that were the best candidate to influence the patients’ follow up, we performed a correlation analysis that identified stage and PreS low/high CTC number to positively and significantly associate with death (R = 0.479, *p* = 0.038 and R = 0.717, *p* = 0.001, respectively), as well as stage and tumor treatment (mitotane only or plus other chemotherapies) with recurrence (R = 0.554, *p* = 0.014 and R = 0.478, *p* = 0.039, respectively). Conversely, no significant correlation was found between the tumor treatment with death, and PreS low/high CTC number with recurrence (not shown).

We analyzed the prognostic power of the CTC number before and after surgery for disease-free survival (DFS) and overall survival (OS) by applying Kaplan–Meier analysis to patients stratified in high and low numbers of CTCs in PreS and PostS blood samples. No significant predictive power for PostS CTC number was found (not shown). Conversely, the 75th percentile cut off value obtained in PreS was able to significantly predict OS (log rank = 0.005, Figure 5A) but not DFS (not shown), confirming the absence of a significant correlation between PreS low/high CTC number and recurrence.

The risk of death (hazard ratio, HR) associated with PreS CTC number was still statistically significant in a multivariate Cox regression analysis after adjusting for age and stage, as shown in Table 2.

Interestingly, according to the Kaplan–Meier analysis, the ability of PreS CTC number to predict the patient survival (log rank = 0.005, χ^2^ = 7.9) was comparable to the metastatic stage IV (log rank = 0.000, χ^2^ = 11.3, Figure 5B), while tumor aggressiveness (dichotomized in two classes of low and high aggressiveness according to stage I–II vs. stage III–IV, respectively) was not a statistically significant parameter (Figure 5C). Finally, in the aggressive stage (stage III–IV), PreS CTC number significantly predicted OS, further refining patients’ stratification (log rank = 0.000, χ^2^ = 12.0, Figure 5D).

## 3. Discussion

Moving further from our previous publication where we showed the presence of CTCs in blood samples from patients with ACC [8] and suggested a diagnostic value for CTCs, in the present paper, we demonstrate the prognostic value of CTC count in the preoperative blood samples of ACC patients. Moreover, our findings indicate that CTCs represent a novel marker obtained through minimally invasive blood sampling to monitor ACC status and evolution before and after surgery, as well as in response to MTT therapy during the patient’s follow-up.

Interestingly, we found CTCs in nearly 68% of PreS blood samples independently from the patients’ stage. The percentage of CTC-positive samples as well as the median number of CTCs decreased significantly within months after surgical removal of the mass. In particular, when patients were stratified on the basis of high and low numbers of CTCs in PreS samples, the PostS CTC number significantly decreased compared to PreS samples. Notably, the number of CTCs in PostS blood samples seemed to reflect the removal of the tumor mass as well as the start of the MTT treatment (two months from surgery). Accordingly, in patients treated post-surgery with MTT only, with no combination with other chemotherapies, a negative correlation between circulating levels of MTT and CTCs was found. Despite the strength of this association, the patients were very few, thus this result needs to be further validated in a larger ACC cohort.

In the present paper, we showed that the tumor recurrence in some representative patients was associated with a rapid increase in CTC number. These findings strongly suggest that CTC count may represent a valid tool for the monitoring of patients during the follow-up. If this result was validated in larger cohort of ACC patients, this might revolutionize the monitoring of patients as well as their quality of life, allowing a strict and sensitive follow-up based only on a simple blood drawing.

The number of cells is so far the only CTC parameter with a recognized validity for both tumor diagnosis and prognosis as well as for tumor monitoring [9,18]. The majority of these studies have been performed in colorectal, breast, lung and prostate cancer, where CTC capture from the blood has been performed by the CellSearch method [19], mainly based on capture by antibodies against cytokeratins—a marker of the epithelial to mesenchymal transition (EMT). However, the EMT phenomenon interests tumors developed in organs of epithelial origin, which is not the case in ACC [7]. In this tumor, the CellSearch method is likely to underestimate the number of CTCs present in the bloodstream [7]. Conversely, in ACC, CTC recovery by size with ScreenCell filters has been demonstrated to work efficiently [8]. The use of ScreenCell filters allows not only the detection of CTCs but also of CTMs and in particular of CAMLs, which is not possible with the use of CellSearch. In fact, in a small number of patients, in addition to CTCs, small clusters of CTCs, named CTMs, were also found entrapped in the filters, along with CAMLs. A statistically significant positive correlation was also found between blood samples with CTMs and circulating CAMLs. Interestingly, CAML presence has also been described in some CTMs with other tumor associated stromal cells [20], although in blood samples from ACC patients we only found CAMLs as single cells. CTMs have been described to detach from the tumor mass as small aggregates of cells, suggesting that they actively migrate from the tumor by a collective migration process and do not associate secondarily in the bloodstream [21]. In these multicellular aggregates, CTCs have been hypothesized to be more resistant to apoptosis and be able to survive also in the blood stream [21,22]. CAMLs are disseminated tumor-associated macrophages (TAMs) with expressing engulfed organ-specific markers. TAMs’ role in cancer progression has been described for the very first time in breast cancer, where they promote angiogenesis, tumor cell migration, metastasis, and immune evasion [23]. CAMLs are highly differentiated giant phagocytic cells of myeloid origin (CD14/CD11c) with large atypical nuclei or multiple individual nuclei, and epithelial cell characteristics, often positive for CD45. The presence of CTMs and CAMLs has been described to be associated with malignancy and poor prognosis in breast cancer [20,24,25,26,27] and other solid tumors [28]. Although found in 36% of our patients and with a significant correlation with CTMs, the small dimension of our cohort, due to the rarity of ACC compared to breast cancer, did not permit the analysis of the prognostic value of either CAMLs or CTMs.

Conversely, we were able to assess the predictive role of single CTC number in both pre-and post-surgery blood samples. In fact, Kaplan–Meier analysis of ACC patients stratified in low and high subgroups based on PreS CTC number demonstrated the ability to significantly predict the overall survival. This relevant finding was obtained by a minimally invasive procedure that does not need tumor removal, which is necessary to obtain the classical tumor pathological parameters. The cut-off value proposed for CTCs in ACC is in line with the one established by the very first paper on breast cancer [11] and with that currently adopted for other solid tumors [29]. Interestingly, although less indicative than the metastatic stage, PreS CTC count was more efficacious in predicting OS than aggressiveness, which became significant only when combined with CTC number. Notably, in our cohort, only one patient (stage IV) did not undergo surgical removal of the ACC mass. This patient was classified in the high CTC number PreS group with a poor prognosis, the filter also showed the presence of two CAMLs. Indeed, the patient died within six months from ACC diagnosis. This patient, who showed bone metastases at (^18^F)-Fluorodeoxyglucose positron emission tomography (^18^FDG-PET) but could not be operated on and classified by classical tumor pathological parameters, is an example of how a simple non-invasive blood drawing aimed at detecting CTC number may assume a valid prognostic value. This type of patient, where staging is difficult as the tumor is not available and classical tumor markers cannot be evaluated, although rare, would greatly benefit from the prognostic power of CTC number, as detected by a simple minimally invasive blood drawing. However, CTC number would also represent a powerful marker for tumor monitoring and its contribution to prognosis in all ACC patients, independently from stage. 

We recognize our study presents some limitations: i. the low number of patients analysed, which is due to the rarity of the tumour; ii. the lack of preS and postS paired samples in some patients, that determined the impossibility to perform a paired analysis; iii. the possible intra-patient variability in the number of CTC found by a single blood sampling; iv. the hetherogeneity of chemotherapy treatment, since some patients received mitotane and others received mitotane plus chemotherapy.

## 4. Materials and Methods

### 4.1. Patients and Ethical Approval

All patients have been monitored for adrenocortical cancer pathology at the University Hospital Azienda Ospedaliera Universitaria (AOU) Careggi. They have been enrolled in the study after giving their written informed consent according to the approval by the local ethical committee (prot. 2011/0020149, Rif CEAVC Em. 2019-201 26/11/2019).

All patients underwent surgical removal of the tumor except one, where ACC diagnosis was made instrumentally by computed tomography (CT)/magnetic resonance imaging (MRI) tumor characteristics. Among operated patients, *n* = 14 presented negative margins (R0) after surgery, while *n* = 4 had positive margins. Tumor samples were formalin-fixed, and paraffin embedded for histologic and immunohistochemical analysis. All patients received MTT therapy (either before or after surgery), of whom, *n* = 9 received MTT only and *n* = 10 MTT plus other chemotherapies (etoposide, doxorubicin and cisplatin (EDP)).

### 4.2. Histologic Analysis and Immunohistochemistry of the Primary Tumour

Histologic diagnosis was performed by the reference pathologist on tumor tissue removed at surgery (*n* = 18). Tumor specimens were evaluated according to the Weiss score system in which the presence of three or more criteria highly correlates with malignant behavior [30,31]. The Ki67 labelling index (Ki67) was evaluated as a proliferation marker to assess ACC prognosis using the anti-human Ki67 antibody (1:40 dilution, clone name MIB-1, Dako, Carpenteria, CA, USA). Ki67 positive nuclei were counted in 1000 tumor cells and Ki67 was expressed as the percentage of labelled cells.

Tumor stage was evaluated according to the revised TMN classification of ACC proposed by the European Network for the Study of Adrenal Tumours (ENS@T) [32].

### 4.3. Blood Sample Collection

For each patient, 6 mL of blood was collected in ethylenediaminetetraacetic acid (EDTA) tubes. Sampling was performed before surgery or at different time points during post-surgical follow-up, with a median (min-max) value of 16 (3–81) months. All blood samples were processed within 3 h after collection [33], generating two filters, and then evaluated for CTC presence.

### 4.4. Mitotane Measurement

Mitotane was measured through the Lysosafe service (HRA Pharma, Châtillon, France) in serum blood samples taken from ACC patients.

### 4.5. CTC Analysis

CTC analysis was performed through three sequential steps consisting of isolation from blood by filtration on ScreenCell^®^ Cyto filtration devices (ScreenCell, Paris, France), followed by CTC identification through validated morphometric criteria [8,10].

#### 4.5.1. CTC Isolation

Blood was filtered by the ScreenCell^®^ Cyto filtration devices (ScreenCell), according to the procedure previously described [8]. Briefly, before filtration, 3 mL blood samples were diluted in a final volume of 7 mL with a specific dilution buffer for fixed cells (ScreenCell^®^ FC dilution buffer, ScreenCell). After filtration, an additional 1 mL of phosphate buffered saline (PBS) was filtered to remove red blood cell (RBC) debris. Filtration was usually completed within approximately 50 s. The filter was then disassembled from the filtration module and allowed to air dry. For each patient’s blood sample, filtration was performed in duplicate.

#### 4.5.2. CTC Staining and Identification

Cytologic study was conducted directly on the filter. The track-etched filters were stained with hematoxylin solution S (Merck KGaA, 64,271 Darmstadt, Germany), applied to the membrane for 1 min, and Shandon eosin Y aqueous (Thermo Electron Corporation, Thermo Fisher Scientific Inc., Waltham, MA, USA) for 45 s. Alternatively, on the second filter obtained from the same 6 mL blood sample, May–Grünwald–Giemsa staining was performed. For microscopic observation, the ScreenCell® Cyto filter was placed on a standard microscopy glass slide and a 7 mm circular cover slip (Menzel-Glaser, Braunschweig, Germany) was laid on the filter with the appropriate mounting medium. CTCs were identified according to the following morphologic criteria: cell size ≥16 μm, nucleocytoplasmic ratio ≥50%, irregular nuclear shape, hyperchromatic nucleus, and basophilic cytoplasm. Under these criteria, red cells and platelets were not entrapped in the filters and leukocytes could be excluded [8]. Blinded cytological analysis was performed under optical microscopy by 2 independent observers (GC and GN) on the 2 differently stained couple of filters (MGG and H&E).

#### 4.5.3. Cytologic Characterization by Immunocytochemistry

To confirm the adrenocortical origin of isolated CTCs, we performed immunostaining for steroidogenic factor-1 (SF-1) of a second filter obtained by the same blood sample, as previously described [8]. Briefly, the filter was incubated for 5 min at room temperature with 70 µL of permeabilizing buffer and subjected to heat-induced epitope retrieval in a bath containing target retrieval solution (S2367, Dako, Glostrup, Denmark) pH 9.0, at 99 °C for 20 min. After washing, the filter was stained with the polyclonal antibody anti-steroidogenic factor 1 (cat #07-618 Upstate, Millipore, Billerica, MA, USA). Staining was achieved by treating the filter with 70 µL EnVision Detection System Peroxidase/ 3.3′-Diaminobenzidine DAB, Rabbit/Mouse (K5007, Dako) for 40 min at room temperature followed by the chromogen 3.3′ diaminobenzidine (Dako) for 10 min at room temperature. Each filter was then placed on paraffin film and the nuclei were slightly counterstained with H&E for 6 min.

### 4.6. Statistical Analysis

Data were expressed as mean ± SD except for CTC number, which was non-normally distributed (as assessed by Kolmogorov–Smirnov’s test) and expressed as median (25–75 percentile). Statistical analysis was performed by SPSS 24.0 (Statistical Package for the Social Sciences, Chicago, IL, USA) for Windows. *p* values of less than 0.05 were considered statistically significant. Correlation analyses were carried out using a χ^2^ test for categorical and Pearson/Spearman test for continuous variables, respectively. For each patient, when multiple samples were available, the median number of CTCs was used for the analysis in both PreS and PostS samples. Comparison between two groups of data was performed using the non-parametric Mann–Whitney U test and Student’s t test for normally distributed variables. Overall survival (OS) and disease-free Survival (DFS) (defined as the probability (ranging from zero to one) that a patient diagnosed with the disease is still alive (OS) or is free from the disease (DFS) at a time point from surgery, for operated patients, or from diagnosis, for not operated patients) and the relative curves were calculated using the Kaplan–Meier method. Differences in OS and DFS between groups of stratification were assessed with log rank and Cox proportional hazards after adjustment for age and stage.

## 5. Conclusions

Though preliminary, this is the first study that demonstrates the predictive power of CTC count in blood liquid biopsy before having access to the tumor analysis through surgery. Our findings need to be further validated in larger cohorts of patients by multicenter studies.

Standardization of studies on CTC evaluation and validation in larger prospective cohorts of ACC patients is mandatory before we can definitely establish the clinical utility of this promising marker in assessing the prognosis, monitoring the disease and guiding the treatment of ACC patients.

## Figures and Tables

**Figure 1 cancers-12-03176-f001:**
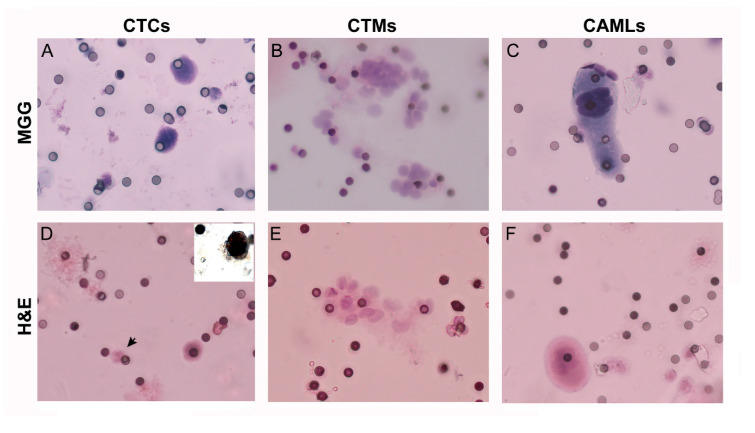
Representative staining with hematoxylin and eosin (H&E) and May–Grünwald–Giemsa (MGG) of circulating cells enriched by ScreenCell filters from blood samples of ACC patients. Circulating tumor cells (CTCs, **A**,**D**), circulating tumor microemboli (CTMs, **B**,**E**), and cancer associated macrophages (CAMLs, **C**,**F**) were separated by ScreenCell devices from blood samples of ACC patients (magnification: 40×) and stained with hematoxylin and eosin (H&E) or May-Grünwald Giemsa (MGG). A polymorphonuclear leukocyte with the typical multilobated nuclei is indicated by the arrow (**D**) for comparison with a CTC. The small pores of 6 µm are visible as small dots non homogenously distributed on the filter membrane. Inset in panel (**D**) shows the nuclear positivity to steroidogenic factor-1 (SF-1) antibody in a representative CTC obtained in a parallel filter from the same blood sample.

**Figure 2 cancers-12-03176-f002:**
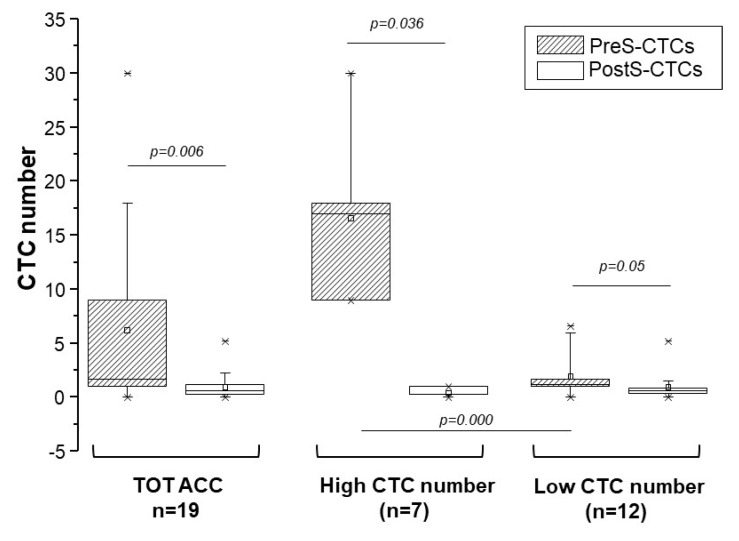
CTC distribution in pre-surgery and post-surgery blood samples in ACC patients. Box charts indicate CTC number distribution between pre-surgery (PreS-CTCs) and post-surgery (PostS-CTCs) blood samples in the entire cohort of ACC patients (TOT ACC), and when the cohort was stratified in low and high CTC number in PreS blood samples classes according to the cut off of 9 CTCs (75th percentile of the non-parametric distributions of PreS CTCs). For each patient, when multiple samples were available, the median number of CTC was used for the analysis; * the lowest and the highest values for each group are indicated. As paired PreS and PostS samples were not available for all patients, differences between the two groups of data were analyzed by the Mann–Whitney U test for independent non-parametric data, the resulting *p* values are indicated.

**Figure 3 cancers-12-03176-f003:**
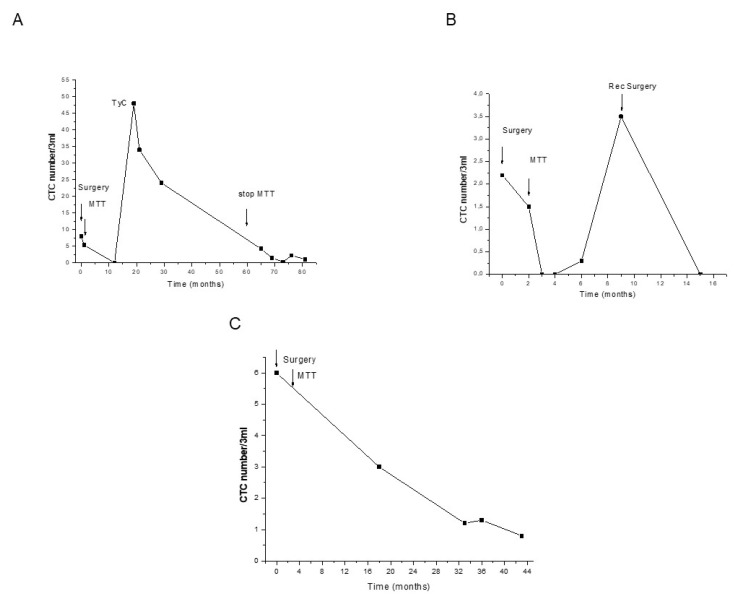
Distribution of CTCs in blood samples during the follow-up of three representative ACC patients. The numbers of CTCs separated and counted on ScreenCell filters in each blood sample at the indicated time of follow-up are represented. Arrows indicate T = 0 corresponding to tumor removal (surgery), the initiation of mitotane (MTT) therapy and the stop of therapy at five years from surgery (60 months). (**A**) patient, stage II, R0; (**B**) patient, stage III, R0, the additional arrow indicates surgery for a retroperitoneal lymph node diagnosed at month 6; (**C**) patient stage III, R0.

**Figure 4 cancers-12-03176-f004:**
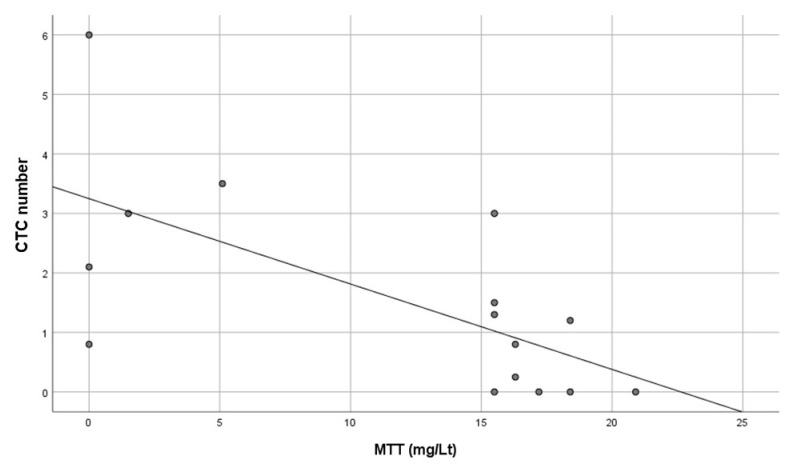
Association between mitotane (MTT) levels and CTC numbers evaluated in the same blood sample in three ACC patients receiving MTT only as post-surgery chemotherapy. A statistically significant negative linear correlation was found between CTC number counted on ScreenCell device (ordinate) and MTT levels (abscissa); R = −0.661, R^2^ = 0.437, *p* = 0.007, *n* = 15.

**Figure 5 cancers-12-03176-f005:**
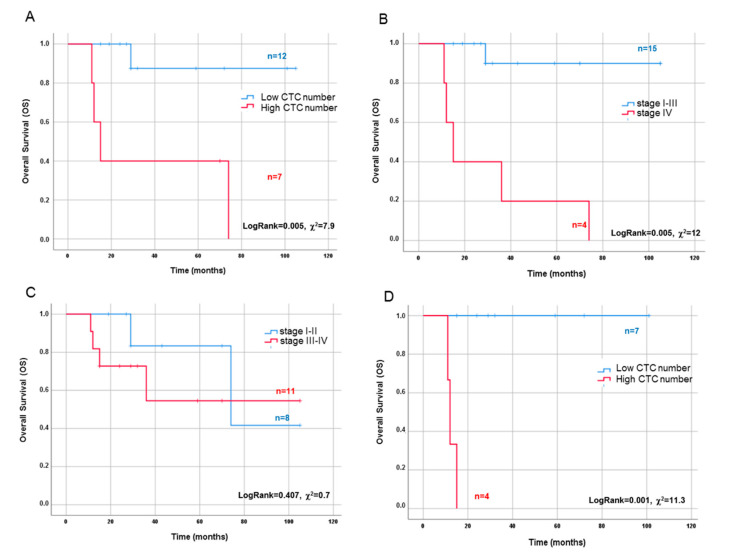
Comparison between the overall survival predictive value of CTC count and stage in ACC patients. Overall survival (OS) Kaplan–Meier curves evaluated in the whole cohort of patients according to stratification in low/high classes of CTC count in PreS samples (cut off value: 9 cells/3 mL) (**A**), in metastatic vs. non-metastatic classes (stage IV vs. I–III) (**B**), and in aggressive vs. non-aggressive classes (stage III–IV vs. I–II) (**C**). Stratification of aggressive stage III–IV patients according to low/high classes of CTC count in PreS samples (**D**). Log rank and χ^2^ values are indicated as well as the number of patients in each class.

**Table 1 cancers-12-03176-t001:** Characteristics of the adrenocortical carcinoma (ACC) patients and of the tumors along with circulating tumor cell (CTC) number in pre-surgery (PreS) and post-surgery (PostS) blood samples. Data from the whole cohort of patients or when patients were stratified in low and high number of CTCs in PreS samples (PreS) using the 75th percentile of PreS CTC distribution as cut off are shown.

	Age (Years)	Sex	Tumor Ø (cm)	Ki67 (%)	Weiss	Stage	Secretion	CTC PreS	CTC PostS
Total cohort (*n* = 19)	53.8 ± 12.5	F:9 (47%)	11.0 ± 4.5	21.4 ± 11.0	6.8 ± 1.2	I: 2 (10%)	NO: 4 (21%)	1.7 [1–9]	0.6 [0.1–1.2]
						II: 6 (32%)	G: 10 (53%)		
		M:10 (53%)				III: 7 (37%)	A: 4 (21%)	*(0–30)*	*(0–5.2)*
						IV: 4 (21%)	M: 1 (5%)		
Low CTC number PreS (*n* = 12)	50.8 ± 9.5	F: 7 (58%)	11.2 ± 4.6	20.4 ± 11.6	6.7 ± 1.2	I: 1 (8%)	NO: 2 (17%)	1.1 [1–2.0]	0.6 [0–0.9]
						II: 4 (33%)	G: 6 (50%)		
		M: 5 (42%)				III: 7 (59%)	A: 3 (25%)	*(0–6.6)*	*(0.1–1.1)*
						IV: 0 (0%)	M: 1 (8%)		
High CTC number PreS (*n* = 7)	57.2 ± 11.8	F: 4 (57%)	9.6 ± 5.4	27.5 ± 9.6	7.8 ± 1.0	I: 1 (14%)	NO: 2 (29%)	17 [9–24]	0.3 [0–0.3]
						II: 2 (29%)	G: 4 (57%)		
		M: 3 (43%)				III: 0 (0%)	A: 1 (14%)	*(9–30)*	*(0–1)*
						IV: 4 (57%)	M: 0 (0%)		

Data are expressed as mean ± SD for parametric (age, tumor diameter, Weiss, Ki67%), or median [25–75 percentiles] for non-parametric continuous variables, and as absolute number and percentage of patients for non-continuous variables (sex, stages, secretion type—G: glucocorticoids, A: androgens, M: mineralocorticoids). Data intervals (min-max) are indicated in italics in brackets. Italics specifically indicates data interval (min-max).

**Table 2 cancers-12-03176-t002:** CTC number in pre-surgery blood samples is the best predictor of death risk in ACC patients. Univariate (unadjusted) and multivariate (adjusted) Cox regression analysis of the risk of death hazard ratio (HR) (95%confidence interval, CI) in univariate and multivariate model for the indicated parameters. Significant *p* values are indicated in italics.

	HR (95%CI) Unadjusted	*p*	HR (95%CI) Adjusted	*p*
Pre-S CTCs	12.0 (1.3–110.0)	*0.026*	22.0 (0.9–540.0)	*0.05*
Age	1.05 (0.99–1.11)	0.120	0.98 (0.86–1.10)	0.675
Stage	4.37 (1.2–15.75)	*0.024*	3.78 (0.7–19.7)	0.114

Italics indicates a statistically significant *p* value compared to non-significant.

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
