# Peer review of "Prognostic and Monitoring Value of Circulating Tumor Cells in Adrenocortical Carcinoma: A Preliminary Monocentric Study"

_cancers, 2020, doi:10.3390/cancers12113176_

Round 1

Reviewer 1 Report

Prognostic and Monitoring Value of Circulating Tumour Cells in Adrenocortical Carcinoma: A Preliminary Monocentric Study

Cantini and colleagues report preliminary results examining circulating tumor cells (CTCs) preoperatively and postoperatively. CTCs are detected in 19 patients and shown to decrease postoperatively. In addition, patients with CTCs greater than the 75th percentile have a worse overall survival (OS), but no disease-free survival (DFS) compared to patients with CTCs less than the 75th percentile. This study is a preliminary case for the use of CTCs as a predictor of survival in a limited cohort of patients with resectable adrenocortical carcinoma (ACC). The findings are interesting, but the cohort is limited and the analysis may benefit from being streamlined.

Introduction, page 3, line 53, It should read management of “advanced adrenocortical carcinoma,” since the poorly effective treatments described are indicated for advanced ACC.

Introduction, page 3, line 55, radical resection is the gold standard for early stage resectable ACC. Please specify as such.

Results, Table 1, line 95, were all the tumors that were functional secreting cortisol? The authors should specify what hormone was being secreted (cortisol, aldosterone, etc.).

Results, Table 1, page 4, line 95, what treatment did these patients receive before and after surgery? This information should be included in the table and should be analyzed by multivariate analysis to see if this was a factor.

Results, page 6, line 133, suggestions and applicability of the findings should be discussed only in the discussion. Please remove.

Results, page 6, line 134, Figure 3 only has three panels (A-C). Please correct.

Results, page 6, line 145, the association between MTT and CTC was assessed for three patients. This data was not interpreted in the discussion. This should be interpreted in the context of ACC. However, the value of this association appears to be limited considering that it was only three patients.

Results, page 7, line 168, the multivariate Cox regression analysis should be listed in a table with all variables utilized.

Results, page 8, line 177, Figure 5, panel C is not listed in the caption.

Results, page 8, line 177, Figure 5, panel C has 11 patients in Stage III-IV group and panel D only has a total of 10 patients when stratifying that group. This discrepancy should be corrected and re-analyzed.

Discussion, page 9, line 198, this sentence is misleading. CTCs were found in nearly 68% of all samples obtained. The percentage of patients with detectable CTCs would be more informative. Such as ten of nineteen patients had detectable CTCs. An analysis of the coefficient of variation would also be helpful for readers to understand the accuracy of the tests.

Discussion, page 10, line 253, although the authors draw a valid point regarding the usefulness of CTCs in predicting patients who may and may not benefit from CTC assessment, this is only one patient, and therefore much of this paragraph is speculative regarding the benefit. In addition, this patient had unresectable ACC and was not a candidate. Patients with unresectable ACC have a worse prognosis than those who are eligible candidates for surgery. This confounds the assessment of utilizing CTC to decide operative management.

Material and Methods, page 11, line 280, how many operations did each patient undergo? What was the patient resection status post-surgery? Were margins positive for any patients? Margin positivity might indicate the reason DFS was not significantly different.

Material and Methods, page 11, line 298, please specify the median time to blood collection and range. It is stated in the results between 3 and 80 months. However, please state the median.

Materials and Methods, page 11, line 310, since it appears multiple samples were collected for each patient, which value was used for CTC in the analysis? The average number of CTCs from all samples? The median? The highest number of CTCs?

Material and Methods, page 11, line 335, please specify how DFS and OS were calculated.

Material and Methods, page 12, line 341-2, were the changes in CTC between pre and post surgery compared using a paired U Mann Whitney’s test or Student T-test? If so, please state. If not, please reanalyze as a paired test.

Minor comments:

Introduction, page 3, line 50, please remove “but.” The addition of the word “but” implies that rare tumors are not commonly aggressive and heterogenous, which reads as an assumption.

Reviewer 2 Report

The article “Prognostic and monitoring value of circulating tumor cells in adrenocortical carcinoma: a preliminary monocentric study” reports preliminary results of a very promising lab tool with clinical utility in terms of prognosis, follow-up and characterization of adrenocortical carcinomas. The technique is fast and non-invasive. The use of size filtration techniques may be a powerful way for detection of the CTC and further characterization.

It is known that liquid biopsy may represent a novel method to get further access to the identification of tumors and their follow-up including prognosis and relapse detection.

In this context CTC reduction after surgery can be used as a valid tool to confirm the success of the resection but also to precociously identify recurrences.

The main limitations of the present study as referred by the authors are the reduced number of patients, which is acceptable in view of the rarity of these carcinomas but constitutes also an opportunity to delineate a multicenter protocol to confirm these preliminary results.

The authors are suggested to clarify some aspects in order to somehow improve the paper a little further:

The main criticism the I can point to this study is somehow a lack of specific identification of the ACC’ CTCs. It is true that there is a significant probability that the patient will only have this tumor and so the CTC be of adrenal origin. However the authors could speculate on the possibility to increase the specificity by the use of IHQ markers that identify adrenal cortex cells (like StAR – see Biomedicines  2020, 8(8), 56; https://doi.org/10.3390/biomedicines8080256 “Incomplete Pattern of Steroidogenic Protein Expression in Functioning Adrenocortical carcinomas”)  and ACC specifically (like IGF2 - IGF2 role in adrenocortical carcinoma biology. Endocrine. 2019 Nov;6 6(2):326-337. doi: 10.1007/s12020-019-02033-5.)

The authors could perhaps explain the differences of this technique in comparison to liquid biopsies in which circulating tumor DNA is analyzed. In terms of specificity, performance and clinical applicability. Were DNA liquid biopsies already done in ACC ?

In general terms it is assumed that the presence of CTC in ACC represents a good marker of the tumor. Is it more a marker that the tumor is metastasizing since cells are spreading through the blood?

And finally some minor remarks:

In the Introduction – line 57 - the expression “patient’s monitoring of the therapy and tumor evolution” looks like that it is the patient that will be doing the monitoring! Perhaps the authors meant “monitoring the therapy of each patient and the tumor evolution”

In the Results, (line 105) it is said that 38% of post-surgery filter readings were still positive. Can the authors confirm this high level of positivity after surgery? It doesn’t seem coherent with the results in Table 1 (CTC present in 0,6%?)

In the discussion (line 205) the authors say CTC numbers correlate with the tumor diameter. However in table 1 it seems to be the opposite (more CTCs -  lesser diameter)

In line 222 it should be written “which is not possible” instead of “which ias not possible”

Round 2

Reviewer 1 Report

Prognostic and Monitoring Value of Circulating Tumour Cells in Adrenocortical Carcinoma: A Preliminary Monocentric Study – R1

Giulia Cantini 2020

Thank you for addressing my comments and updating the manuscript.

General comments:

Results, page 3, thank you to the authors for analyzing the margin status. Please include these findings in the Results section. Was margin status associated with overall survival?

Results, page 3, line 99, Table 1. Please add the margin and chemotherapy status to the table. I think this would allow quick reference to the clinical characteristics.

Results, page 6, line 177, would consider using a different phrase other than ‘tumour treatment,’ since surgery is a ‘tumour treatment.’ Would consider using ‘medical treatment’ or ‘chemotherapy’ for clarity.

Results, page 7, line 190, would consider including all variables that were tested in the unadjusted model such as ‘tumor treatment,’ ‘Ki-67,’ ‘Weiss,’ etc. and then leave the variables used in the multivariate analysis.

Discussion, page 9, line 279, the major limitation is number of patients as the authors correctly point out, albeit adrenocortical carcinoma is a rare cancer. The authors should point out other limitations such as not always having pre- and post-samples to compare for a paired analysis, and the heterogeneity of pre- and post-chemotherapy treatment, since some patients received mitotane and others received mitotane plus chemotherapy.

Minor comments:

Results, page 6, line 178, please remove the word “Conversely,” since the sentence is stating a finding and not an opposing finding. The sentence reads better as “No significant correlation…”

Results, page 7, line 187, typo ‘staticallsignificant’ should read ‘statistically significant.’
